# *CLIC4* Is a New Biomarker for Glioma Prognosis

**DOI:** 10.3390/biomedicines12112579

**Published:** 2024-11-11

**Authors:** Zhichun Liu, Junhui Liu, Zhibiao Chen, Xiaonan Zhu, Rui Ding, Shulan Huang, Haitao Xu

**Affiliations:** Department of Neurosurgery, Renmin Hospital of Wuhan University, Wuhan 430060, China; l19921874784@163.com (Z.L.); doctorhsdqe@whu.edu.cn (J.L.); chzbiao@126.com (Z.C.); zhuxiaonan@gmail.com (X.Z.); rui-ding@hotmail.com (R.D.)

**Keywords:** glioma, *CLIC4*, immune infiltration, ECM

## Abstract

Background: Chloride Intracellular Channel 4 (*CLIC4*) plays a versatile role in cellular functions beyond its role in primary chloride ion transport. Notably, many studies found an association between *CLIC4* expression and cancers. However, the correlation between *CLIC4* and glioma remains to be uncovered. Methods: A total of 3162 samples from nine public datasets were analyzed to reveal the relationship between *CLIC4* expression and glioma malignancy or prognosis. Immunohistochemistry (IHC) staining was performed to examine the results in an in-house cohort. A nomogram model was constructed to predict the prognosis. Functional enrichment analysis was employed to find *CLIC4*-associated differentially expressed genes in glioma. Immune infiltration analysis, correlation analysis, and IHC staining were employed, aiming to examine the correlation between *CLIC4* expression, immune cell infiltration, and ECM (extracellular matrix)-related genes. Results: The expression level of *CLIC4* was correlated with the malignancy of glioma and the prognosis of patients. More aggressive gliomas and mesenchymal GBM are associated with a high expression of *CLIC4*. Gliomas with IDH mutation or 1p19q codeletion express a low level of *CLIC4*, and a high expression of *CLIC4* correlates with poor prognosis. The nomogram model shows a good predictive performance. The DEGs (differentially expressed genes) in gliomas with high and low *CLIC4* expression are enriched in extracellular matrix and immune functions. On the one hand, gliomas with high *CLIC4* expression have a greater presence of macrophages, neutrophils, and eosinophils; on the other hand, a high *CLIC4* expression in gliomas is positively associated with ECM-related genes. Conclusions: Compared to glioma cells with low *CLIC4* expression, gliomas with high *CLIC4* expression exhibit greater malignancy and poorer prognosis. Our findings indicate that a high level of *CLIC4* correlates with high expression of ECM-related genes and the infiltration of macrophages, neutrophils, and eosinophils within glioma tissues.

## 1. Introduction

Gliomas are tumors that originate from glial cells in the brain and spinal cord. They make up about 30% of all brain and central nervous system tumors and 80% of malignant brain tumors [1]. These tumors are categorized based on the type of glial cell they arise from, including astrocytomas, oligodendrogliomas, and ependymomas.

Gliomas are further classified by their aggressiveness. Lower-grade gliomas (grades I and II) tend to grow more slowly and are less aggressive, while higher-grade gliomas (grades III and IV), such as glioblastoma multiforme (GBM), are more rapidly growing and aggressive [2].

The prognosis of gliomas varies significantly based on molecular characteristics and subtypes. Low-grade gliomas gener.ally have a favorable prognosis, especially those with IDH mutations and 1p19q codeletion. In contrast, high-grade glioblastomas (GBMs), particularly the mesenchymal subtype, are associated with poor outcomes due to their high invasiveness and resistance to therapy, resulting in markedly reduced survival times. IDH mutations and 1p19q codeletion are key genetic markers in glioma classification. IDH mutations, common in lower-grade gliomas, correlate with less aggressive tumor behavior and improved prognosis. The 1p19q codeletion, characteristic of oligodendrogliomas, further indicates favorable treatment response and survival. Together, IDH mutation and 1p19q codeletion define a glioma subtype with superior clinical outcomes, refining prognosis and guiding therapy [3]. Verhaak et al. previously classified glioblastoma multiforme (GBM) into four sub-types—classical, mesenchymal, proneural, and neuronal—based on distinct genomic expression profiles. GBM cells with a mesenchymal phenotype exhibited higher levels of cell motility-related proteins, along with enhanced migration and invasion, compared to the classical and proneural subtypes [4].

Chloride Intracellular Channel 4 (*CLIC4*) plays multiple roles in cellular functions beyond its primary task of transporting chloride ions. It exists in both soluble and membrane-bound forms, with its expression regulated by stress and growth signals. *CLIC4* influences the cell cycle, apoptosis, and cell survival, adapting its functions to specific cellular environments [5]. In addition, *CLIC4* responds to cellular stress and contributes to both tumor suppression and cancer progression, with its expression levels associated with tumor growth and patient prognosis. Its role in angiogenesis—essential for tumor development and wound healing—highlights *CLIC4*’s significant impact on cancer research. Deng YJ et al. found that a high expression of *CLIC4* may contribute to the aggressive abilities of colorectal cancer [6].

In this study, we analyzed data from 3162 samples obtained from various public datasets, including the Cancer Genome Atlas-GBMLGG (699 samples) [7], Chinese Glioma Genome Atlas-mRNAseq_693# (1018 samples) [8], Gill (92 samples) [9], Gravendeel (284 samples) [10], Murat (84 samples) [11], Rembrandt (580 samples) [12], LeeY (191 samples) [13], GSE7696 (84 samples), and GSE50161 (130 samples). Our goal was to investigate the expression levels of *CLIC4* and its association with malignancy and prognosis in gliomas. To validate these findings, we performed immunohistochemical (IHC) staining on 56 paraffin-embedded glioma samples and 12 non-tumor brain tissues. Our results suggest that *CLIC4* could serve as a novel prognostic biomarker for gliomas, with its expression linked to ECM-related genes and immune cell infiltration within tumors.

## 2. Methods and Materials

### 2.1. Glioma Tissues

We utilized a tissue microarray consisting of 56 paraffin-embedded glioma samples and 12 non-tumor brain tissues. All specimens were obtained from patients admitted to the Department of Neurosurgery at Renmin Hospital of Wuhan University from January 2016 to March 2018. The study received approval from the Institutional Ethics Committee of Renmin Hospital of Wuhan University [approval number: 2012LKSZ (010) H].

### 2.2. Data Collection

The mRNA expression profiles and clinical data for glioma patients were sourced from the GlioVis platform (http://gliovis.bioinfo.cnio.es, accessed on 23 November 2023) and the Gene Expression Omnibus (GEO) (https://www.ncbi.nlm.nih.gov/geo/, accessed on 23 November 2023). Nine datasets were utilized in the analysis, including TCGA, CCGA, Granvendeel, Rembrandt, Gill, Murat, LeeY, GSE7696, and GSE50161. Graphpad Prism10 (GraphPad Inc., San Diego, CA, USA) was employed to analyze six datasets including Granvendeel, Rembrandt, Gill, Murat, GSE7696, and GSE50161 to show the different expressions of *CLIC4* in NBT and glioma.

### 2.3. Immunohistochemical Staining

Immunohistochemical (IHC) staining was performed on formalin-fixed paraffin-embedded (FFPE) human glioma and normal brain tissue samples. The paraffin-embedded tissue microarray was heated at 60 °C for 90 min. The slides were then rehydrated in xylene (three 5 min doses) and graded ethanol solutions (100%, 95%, and 75%). Following three rinses in PBS, 3% hydrogen peroxide (H_2_O_2_) was applied to the slides for 10 min at room temperature.

Incubation at 95 °C for 10 min with an antigen retrieval solution and natural cooling of the slides followed. After permeabilization with Triton-PBS (100×) and blocking with 1% BSA, the slides were incubated with the primary antibody (CLIC4, No. 12298-2-AP, 1:200, Proteintech Wuhan, Hubei, China; CD68, No. 28058-1-AP, 1:2000, Proteintech; VIM, No. 22031-1-AP, 1:200, Proteintech; SNAI1, No. 13099-1-AP, 1:200, Proteintech; MMP1, 10371-2-AP, 1:400, Proteintech) and subsequently with an HRP-conjugated goat anti-rabbit IgG secondary antibody (No. 15015, 1:500, Proteintech). Cell nuclei were stained with 3,3-diaminobenzidine (DAB) and protected from light for 10 min at room temperature. Finally, the slides were mounted with neutral balsam and scanned using an Olympus BX40 microscope (Olympus Corporation, Tokyo, Japan).

### 2.4. IHC Evaluation

In this study, we used ImageJ 1.54g (http://imagej.net), a widely used open-source software, to visualize, quantify, and validate the IHC staining results of CLIC4. To enhance accuracy, we employed an interactive tool in ImageJ (http://imagej.net/ij/plugins/ihc-toolbox/index.html, accessed on 5 December 2023), which automatically categorizes pixels in digital slides into positively and negatively stained groups.

A thresholding tool was applied to set a threshold for staining evaluation. Once the thresholds were established, the software calculated the area and integrated density of the positively stained regions. The average optical density was determined by dividing the integrated density by the area. Based on the median value of the mean optical density, glioma and GBM patients were classified into high- and low-expression groups.

### 2.5. Correlation Analysis Between CLIC4 and Mesenchymal Cell-Associated or EMC-Associated Genes

The relationship between CLIC4 expression levels and mesenchymal cell-associated genes (including *SNAI1*, *VIM*, *TWIST1*, *ZEB1*, *CD44*, *CHI3L1*, *RELB*, and *TNFRSF1A*), as well as ECM-associated genes (including *VIM*, *SNAI1*, *MMP1*, *MMP9*, *SNAI2*, *COL1A1*, *COL3A1*, *FN1*, *ITGB1*, *LAMC1*, *HAS2* and *TNC*), was examined in TCGA glioma samples. Spearman’s correlation analysis was conducted using the “ggplot2” (v3.3.3) R package. Statistically significant correlations were those with a *p*-value less than 0.05.

### 2.6. Evaluation of the Prognostic Significance of CLIC4 in Glioma

Survival data of glioma patients from the TCGA database were analyzed using the “survival” (v3.2-10) R package for statistical analysis and the “survminer” (v0.4.9) R package for visualization. Kaplan–Meier survival curve analysis, along with univariate and multivariate Cox regression analyses, was conducted to assess survival outcomes based on *CLIC4* expression levels. Additionally, a nomogram model was developed using the “ggplot2” (v3.3.3) R package to evaluate the prognostic significance of *CLIC4* expression in glioma.

### 2.7. Enrichment Analysis

The “org.Hs.eg.db” (v3.10.0) R package was used to convert entrez IDs to gene symbols. Functional annotation and Gene Set Enrichment Analysis of differentially expressed genes were performed using the “ClusterProfiler” (v3.14.3) R package. For GSEA, curated reference gene sets from the MgDB file (c2.cp.v7.2.symbols.gmt) were selected. Enriched clusters were identified based on a false discovery rate of less than 0.25 and an adjusted *p*-value of less than 0.05.

### 2.8. Correlation Analysis Between CLIC4 Expression Levels, Immune Cell Infiltration in Glioma

The ssGSEA algorithm from the “GSVA” (v1.34.0) R package was used to assess the infiltration of 24 immune cell types. Spearman’s correlation analysis was performed to investigate the relationship between *CLIC4* expression levels and immune cell infiltration. A correlation chord diagram was then used to visualize the association between *CLIC4* expression and immune cell markers.

## 3. Results

### 3.1. Transcription Level of CLIC4 in Gliomas of Public Datasets

To investigate *CLIC4* gene expression in gliomas, six public datasets were analyzed, including Gill, Granvendeel, Murat, and Rembrandt from GlioVis, as well as GSE7696 and GSE50161 from the GEO database. The results indicated a significant upregulation of *CLIC4* in gliomas (Figure 1A).

Next, we examined the correlation between *CLIC4* and glioma grading. Four public datasets were used to assess *CLIC4* expression across different glioma grades. The results revealed that *CLIC4* expression was significantly higher in glioblastoma multiforme (GBM) compared to lower-grade gliomas (WHO II-III) (Figure 1B).

IDH1/2 mutations and 1p19q codeletion are two critical mutations in gliomas, often associated with favorable patient prognosis. To explore the relationship between *CLIC4* expression and these mutations, we divided lower-grade glioma (LGG) patients into three groups: Group I (IDH mutation with 1p19q codeletion), Group II (IDH mutation without 1p19q codeletion), and Group III (IDH wild-type). A fourth group consisted of GBM patients with IDH wild-type tumors. An analysis of two public datasets (TCGA and CGGA) showed that Groups I and II had lower CLIC4 expression compared to Groups III and IV (Figure 1C). Additionally, patients with 1p19q codeletion exhibited lower *CLIC4* expression than those in Group II.

### 3.2. Transcription Level of CLIC4 in Different Subtypes of GBM

GBM cells with a mesenchymal phenotype are highly migratory and invasive, driving the aggressive behavior of glioblastoma. This subtype shows elevated expression of epithelial-to-mesenchymal transition (EMT) genes, enhancing motility and apoptosis resistance. In this study, five public datasets (TCGA, CGGA, Rembrandt, Gill, and LeeY) were analyzed, revealing that *CLIC4* expression was significantly higher in mesenchymal GBM compared to proneural GBM. Additionally, in the TCGA dataset, *CLIC4* expression was also significantly elevated in mesenchymal GBM compared to the classical subtype (Figure 2A).

Spearman correlation analysis showed that *CLIC4* was closely associated with several epithelial–mesenchymal transition (EMT)-related markers (*SNAI1*, *VIM*, *TWIST1*) and mesenchymal cell-associated genes (*CD44*, *CHI3L1*, *RELB*, and *TNFRSF1A*), except for *ZEB1* [14] (Figure 2B). These findings suggest that *CLIC4* is predominantly expressed in the mesenchymal subtype of GBM.

### 3.3. Transcription Level of CLIC4 in Gliomas of Clinic Samples

The analysis above suggests that the transcription level of *CLIC4* is related to glioma malignancy and patient prognosis. To validate this, we performed immunohistochemical (IHC) staining on 56 paraffin-embedded glioma samples and 12 non-tumor brain tissues. The results showed that glioma cells exhibited higher levels of CLIC4 compared to normal brain tissue (NBT), with glioblastoma (GBM) displaying significantly higher CLIC4 expression than low-grade gliomas (LGG) (Figure 3A,B).

Moreover, glioma samples from patients with IDH wild-type tumors showed a higher percentage of positively stained areas compared to those with IDH mutations (Figure 3C).

### 3.4. CLIC4 Predict Prognosis in Glioma

After establishing a strong correlation between *CLIC4* expression and glioma malignancy, we further assessed its prognostic value. Survival analysis revealed that, across all glioma grades, *CLIC4*-expressed patients were more likely to have a poor overall survival (OS) percentage in the TCGA, CGGA, Rembrandt, and Gravendeel datasets.

For patients with low-grade gliomas, those with high *CLIC4* expression also had a lower OS percentage than patients with low *CLIC4* expression in the TCGA, CGGA, and Rembrandt datasets. Similarly, in glioblastoma patients, high *CLIC4* expression was associated with a lower OS percentage compared to low *CLIC4* expression in the TCGA, CGGA, and Gravendeel datasets (Figure 4).

### 3.5. The Construction of Nomogram Prediction Model

*CLIC4* expression may be linked to patient prognosis. To explore this further, we constructed a nomogram prediction model. A univariate analysis identified age, WHO classification, IDH status, and *CLIC4* expression level as significant factors affecting the survival of glioma patients. A multivariate analysis confirmed that these factors were independent prognostic indicators (Table 1). Patients older than 60 years had a higher risk of disease-related death compared to those 60 years or younger (*p* = 0.004). Additionally, patients with WHO class IV tumors had an increased risk of death compared to those with WHO class II or III tumors (*p* < 0.001). IDH mutations were associated with a decreased risk of disease-related death compared to the IDH wild-type (*p* < 0.001), and patients with high CLIC4 expression had a higher risk of death compared to those with low *CLIC4* expression (*p* = 0.004).

Using these four independent prognostic factors—age, WHO classification, IDH status, and *CLIC4* expression level—we developed a nomogram to predict 1-, 3-, and 5-year overall survival in glioma patients. The total score was calculated by summing the scores for each factor projected onto the scale, and the corresponding survival rates were determined. A higher total score indicated a poorer overall survival prognosis (Figure 5).

A calibration curve analysis showed that the nomogram closely aligned with the ideal curve for predicting 1-, 3-, and 5-year survival rates, suggesting that the model has strong predictive accuracy (Figure 6).

### 3.6. Functional Enrichment Analysis of CLIC4-Associated Differentially Expressed Genes in Glioma

Glioma patient data from TCGA were divided into high- and low-expression groups based on the median *CLIC4* expression levels, and differentially expressed genes between the two groups were analyzed using GO enrichment analysis. The results indicated that *CLIC4* was primarily enriched in pathways related to the positive regulation of cell adhesion, cytokine production, and the collagen-containing extracellular matrix (Figure 7A).

Subsequently, Gene Set Enrichment Analysis (GSEA) was performed for *CLIC4* in glioma. The *CLIC4* high-expression group showed significant enrichment in pathways such as cytokine signaling in the immune system and the innate immune response. Additionally, pathways related to the extracellular matrix, including matrisome, ECM proteoglycans, extracellular matrix organization, and degradation of the extracellular matrix, were also highly enriched in the high-expression group (Figure 7B). The top 5 of the most upregulated are NABA_MATRISOME (NES = 3.29, *p* = 0.001), REACTOME_CYTOKINE_SIGNALING_IN_IMMUNE_SYSTEM (NES = 3.19, *p* = 0.001), KEGG_CYTOKINE_CYTOKINE_RECEPTOR_INTERACTION (NES = 3.15, *p* = 0.001), WP_OVERVIEW_OF_PROINFLAMMATORY_AND_PROFIBROTIC_MEDIATORS (NES = 3.10, *p* = 0.001), and REACTOME_SIGNALING_BY_INTERLEUKINS (NES = 3.06, *p* = 0.001). The top 5 of the most downregulated are REACTOME_NEURONAL_SYSTEM (NES = −5.16, *p* = 0.004), REACTOME_TRANSMISSION_ACROSS_CHEMICAL_SYNAPSES (NES = −5.09, *p* = 0.004), WP_SYNAPTIC_VESICLE_PATHWAY (NES = −3.51, *p* = 0.003), KEGG_NEUROACTIVE_LIGAND_RECEPTOR_INTERACTION (NES = −3.44, *p* = 0.004), and REACTOME_NEUROTRANSMITTER_RECEPTORS_AND_POSTSYNAPTIC_SIGNAL_TRANSMISSION (NES = −3.20, *p* = 0.003).

These findings suggest that *CLIC4*’s role in glioma may be closely linked to the extracellular matrix and immune cell infiltration.

### 3.7. CLIC4 Expression Levels Correlate with the Immune Infiltration in the Glioma Tissues

The infiltration status of 24 immune cell types in glioma tissues was evaluated using ssGSEA. Spearman’s correlation analysis was then used to assess the association between *CLIC4* expression and immune cell infiltration. *CLIC4* expression showed a positive correlation with macrophages, neutrophils, and eosinophils, and a negative correlation with plasmacytoid dendritic cells (pDCs), NK CD56bright cells, and regulatory T cells (Tregs) (Figure 8A). The tumor infiltration levels of these immune cells (macrophages, neutrophils, eosinophils, pDCs, NK CD56bright cells, and Tregs) (Figure 8B) were consistent with Spearman’s analysis results.

Next, we evaluated the correlation between *CLIC4* expression and immune cell markers. *CLIC4* expression levels were positively correlated with specific markers of macrophages (*CD80*, *CD86*, *MRC1*), neutrophils (*FCGR3A*, *FCGR3B*), eosinophils (*SIGLEC8*, *IL3RA*), T cells (*CD3D*, *CD4*), CD8+ T cells (*CD8A*, *CD8D*), NK cells (*NCAM1*), and B cells (*CD19*) in glioma (Figure 8C). Additionally, immunohistochemical (IHC) analysis revealed that *CD68*, a macrophage marker, was significantly elevated in glioma tissues with high CLIC4 expression (Figure 9A).

### 3.8. CLIC4 Expression Levels Correlate with the ECM-Related Genes in the Glioma Tissues

The results above suggest that *CLIC4* plays a critical role in glioma progression and could serve as a potential biomarker for predicting prognosis. An enrichment analysis also indicated that *CLIC4* is associated with the tumor microenvironment, particularly the extracellular matrix (ECM). To further explore this, we conducted a correlation analysis between *CLIC4* and ECM-related genes. The analysis revealed positive associations between *CLIC4* and *VIM* (r = 0.762, *p* < 0.001), *MMP1* (r = 0.406, *p* < 0.001), *MMP9* (r = 0.516, *p* < 0.001), *SNAI1* (r = 0.507, *p* < 0.001), *SNAI2* (r = 0.495, *p* < 0.001), *COL1A1* (r = 0.631, *p* < 0.001), *COL3A1* (r = 0.631, *p* < 0.001), *FN1* (0.720, *p* < 0.001), *ITGB1* (r = 0.796, *p* < 0.001), *LAMC1* (r = 0.743, *p* < 0.001), *HAS2* (r = 0.666, *p* < 0.001), and *TNC* (r = 0.782, *p* < 0.001) (Figure 10).

Consistent with these findings, an immunohistochemical (IHC) analysis showed that the expression of VIM (Figure 9B), SNAI1 (Figure 9C), and MMP1 (Figure 9D) was significantly higher in glioma tissues with elevated CLIC4 expression.

## 4. Discussion

Recent studies have shown that the *CLIC4* gene is closely related to tumor progression. However, the role of the *CLIC4* gene in glioma remains to be elucidated [5].In this study, we observed high *CLIC4* expression in glioma tissues across six public datasets. *CLIC4* was found to be overexpressed in high-grade gliomas compared to WHO grade II and III gliomas, a finding that was also confirmed by immunohistochemical (IHC) staining. These results suggest that *CLIC4* may play a role in glioma progression. Previous studies have explored the expression and regulation of *CLIC4* in cancer cells. Suh and colleagues proposed that reducing *CLIC4* expression could be a potential therapeutic target in oncology [15]. Another study showed that while *CLIC4* deletion is common in tumor cells, its increased expression in the tumor stroma is associated with malignant progression in various cancers. This implies that reactivating or restoring *CLIC4* expression in tumor cells, or reducing its expression in the tumor stroma, may offer novel strategies for inhibiting tumor growth [16]. *CLIC4* has also been reported to interact with *TGF-β* signaling, enhancing the pathway by inhibiting dephosphorylation of phospho-Smad signaling [17]. Although the precise mechanism by which *CLIC4* functions as a tumor suppressor or matrix activator remains unclear, increasing evidence suggests its clinical relevance.

IDH is a key enzyme in cellular metabolism, catalyzing the conversion of isocitrate to α-ketoglutarate. IDH mutations are found not only in gliomas but also in other tumors, such as cholangiocarcinoma. In cholangiocarcinoma, the prognostic impact of IDH mutations is more complex and may be related to factors such as tumor differentiation and mutation type, whereas in gliomas, IDH mutations are typically associated with a better prognosis [18]. Patients with IDH-mutant gliomas have a longer median survival compared to those with IDH wild-type gliomas, a key prognostic factor. In 2008, Parsons et al. reported that patients with IDH-mutant glioblastomas had a median survival of 31 months, compared to 15 months for those with IDH wild-type tumors [19]. The prognosis of glioma patients is more strongly determined by IDH mutation status than by histologic grade. Studies have shown that patients with WHO grade III or IV IDH-mutant gliomas have better outcomes than those with IDH wild-type tumors [20]. Additionally, patients with WHO grade III IDH-mutant gliomas have a median survival of 65 months, compared to 20 months for those with IDH wild-type tumors [21]. Among IDH-mutant gliomas, oligodendrogliomas with 1p/19q codeletion and TERT promoter mutations have the best prognosis, with a median survival of 8 years [22]. As mentioned above, GBM cells with a mesenchymal phenotype are linked to poor prognosis due to heightened invasiveness and treatment resistance. This subtype exhibits elevated epithelial-to-mesenchymal transition (EMT) markers, enhancing cell migration and invasion. Consequently, mesenchymal GBM shows high recurrence rates and limited response to conventional therapies like chemotherapy and radiotherapy, leading to shorter survival compared to other GBM subtypes. In our study, patients with IDH mutations and 1p/19q deletions exhibited lower *CLIC4* expression, while *CLIC4* overexpression was observed in the mesenchymal subtype, suggesting that lower *CLIC4* expression correlates with better prognosis. Kaplan–Meier survival curves confirmed that lower *CLIC4* expression is associated with improved outcomes. We developed a nomogram model incorporating *CLIC4* and four other independent prognostic factors, which demonstrated strong predictive performance. In fact, the recent literature has revealed that *CLIC4* may be associated with poor tumor prognosis; however, our study is the first to investigate the role of *CLIC4* in glioma prognosis [6].

Since *CLIC4* expression levels vary across different tumors, either increasing or decreasing, the precise mechanism by which *CLIC4* acts as a tumor suppressor or promoter remains unclear. However, it is noteworthy in numerous studies that *CLIC4* may influence tumor growth and development through the tumor stroma.

The role of *CLIC4* in tumor stroma was initially identified by Ronnov-Jessen and colleagues, who observed that *TGF-β* exposure significantly upregulated *CLIC4* expression in the myofibroblast stroma of breast cancer, suggesting its involvement in stromal remodeling during cancer progression [23]. This early discovery provided a foundation for understanding *CLIC4*’s regulatory effects within the tumor microenvironment. Building on this groundwork, Yao et al. explored how *TGF-β1* drives fibroblast-to-myofibroblast differentiation in ovarian cancer, linking this transformation to an upregulated ROS-*CLIC4* pathway. They found that this pathway is essential in enabling fibroblasts to shift phenotypes under oxidative stress, a process frequently activated within the tumor stroma [24]. Further insights emerged from Shukla et al., who, using a skin cancer model, demonstrated that *TGF-β*-driven activation of *p-38*, coupled with blocked dephosphorylation of phospho-p-38, promotes myofibroblast conversion. This mechanism closely resembles the activation pathway of p-Smad signaling [25]. These combined findings indicate that *CLIC4* may be a key player in stromal remodeling contributing to a supportive tumor microenvironment.

Similar to the findings in the above studies, ECM components also emerged as significant in our enrichment analysis. The ECM, consisting of collagen, non-collagenous proteins, elastin, proteoglycans, and glycosaminoglycans, plays a critical role in regulating cellular activities such as growth, migration, and metabolism [26]. The tumor-associated ECM gene is considered a key driver of glioma invasion [27]. Function enrichment analyses revealed that *CLIC4* is associated with various immune processes. Immune infiltration analysis showed that macrophages and other innate immune cells were predominant in gliomas with high *CLIC4* expression. While the role of innate immunity in cancer has been well studied, recent research highlights novel aspects of its involvement in tumor initiation and progression. For instance, abnormal cell proliferation and stress associated with carcinogenesis can trigger the release of danger-associated molecular patterns (DAMPs) [28], activating innate immune pathways to eliminate transformed cells [29], This process promotes the recruitment and expansion of tumor-specific CD8+ T cells, potentially enhancing treatment responses and patient outcomes [30,31,32]. However, unexpected results emerged in the immune analysis. *CLIC4* expression levels were negatively correlated with pDCs and Treg cells, both of which primarily exhibit pro-tumor effects. Studies on *CLIC4* and immune cell infiltration are limited, and the mechanisms behind this phenomenon remain unclear.

Although this study revealed the correlation between *CLIC4* expression levels and ECM-related genes as well as immune cell infiltration in gliomas, the specific mechanisms require further clarification.

## 5. Conclusions

In comparison to glioma cells with lower *CLIC4* expression, those exhibiting elevated *CLIC4* levels show increased malignancy and a worse prognosis. Our results reveal that heightened *CLIC4* expression is associated with elevated ECM-related gene activity and the presence of macrophages, neutrophils, and eosinophils infiltrating glioma tissues.

## Figures and Tables

**Figure 1 biomedicines-12-02579-f001:**
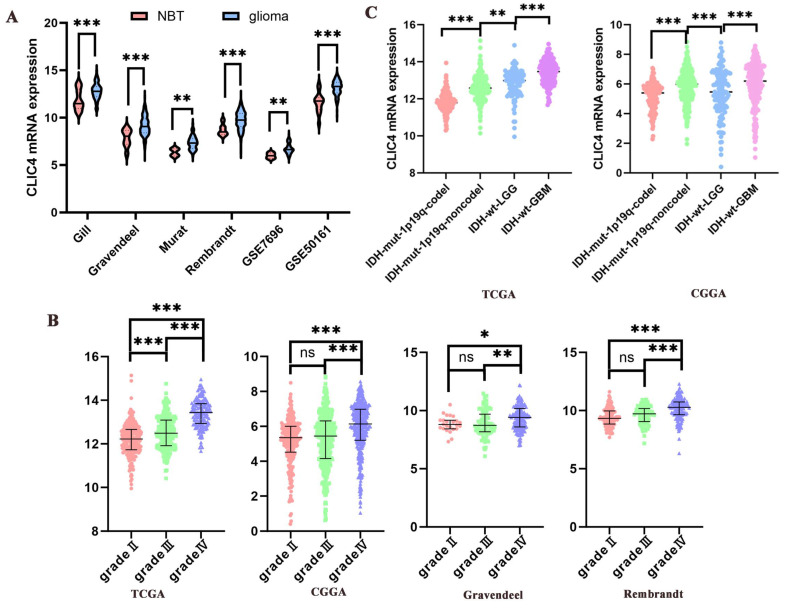
Transcription level of *CLIC4* in gliomas of public datasets. (**A**) *CLIC4* mRNA data of glioma tissues and normal brain tissues from six public datasets. (**B**) *CLIC4* mRNA data of glioma tissues from four public datasets. Each dataset was divided into three groups according to WHO grading. (**C**) *CLIC4* mRNA data of glioma tissues from two public datasets. Each one was divided into four groups according to IDH statue and 1p19q deletion. ns > 0.05, * *p* ≤ 0.05, ** *p* < 0.01, *** *p* < 0.001. NBT means normal brain tissue.

**Figure 2 biomedicines-12-02579-f002:**
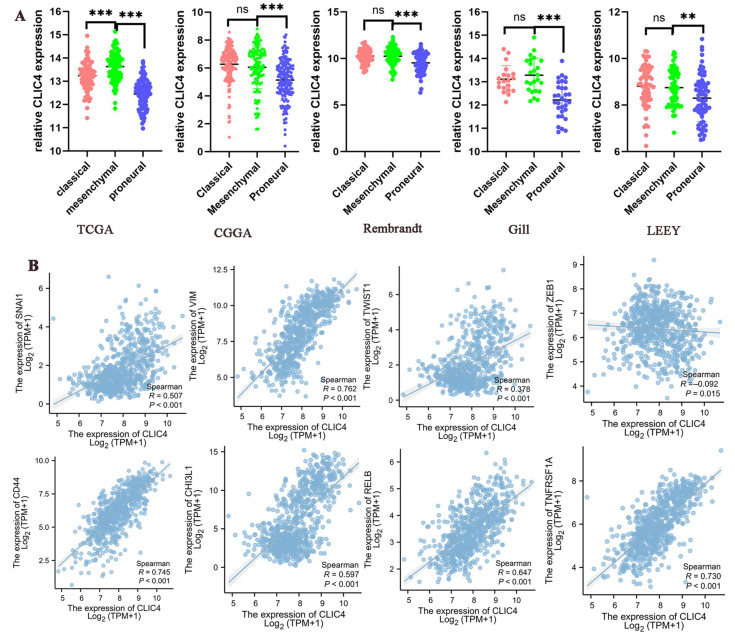
Transcription level of *CLIC4* in different subtypes of GBM. (**A**) *CLIC4* mRNA in different subtypes of GBM from five datasets. (**B**) The correlation between CLIC4 expression and mesenchymal-related genes. ns > 0.05, ** *p* < 0.01, *** *p* < 0.001.

**Figure 3 biomedicines-12-02579-f003:**
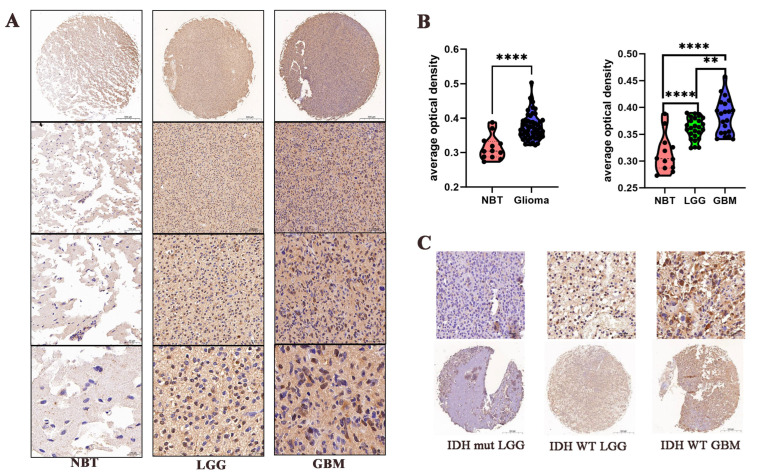
Transcription level of CLIC4 in gliomas of clinic samples. (**A**) Representative IHC images of CLIC4 in normal brain tissue, LGG, and GBM. Scale bars were 500 µm, 100 µm, 50 µm, and 20 µm. (**B**) The average optical density of IHC staining. (**C**) Representative IHC images in IDH mut LGG, IDH WT LGG, and IDH WT GBM. Scale bars were 50 µm and 500 µm. ** *p* < 0.01, **** *p* < 0.0001.

**Figure 4 biomedicines-12-02579-f004:**
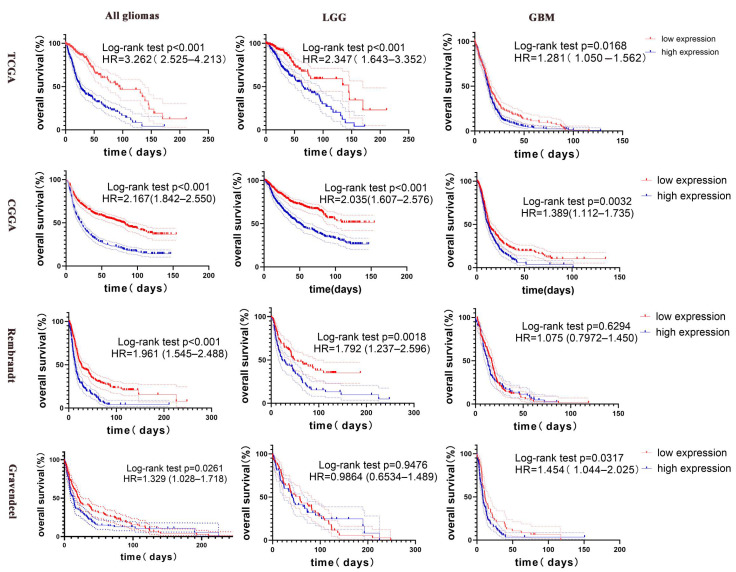
Relationship between *CLIC4* mRNA expression and prognosis in patients with glioma. Differences in survival of different *CLIC4* mRNA expression levels were analyzed by Kaplan–Meier analysis. HR, hazard ratio.

**Figure 5 biomedicines-12-02579-f005:**
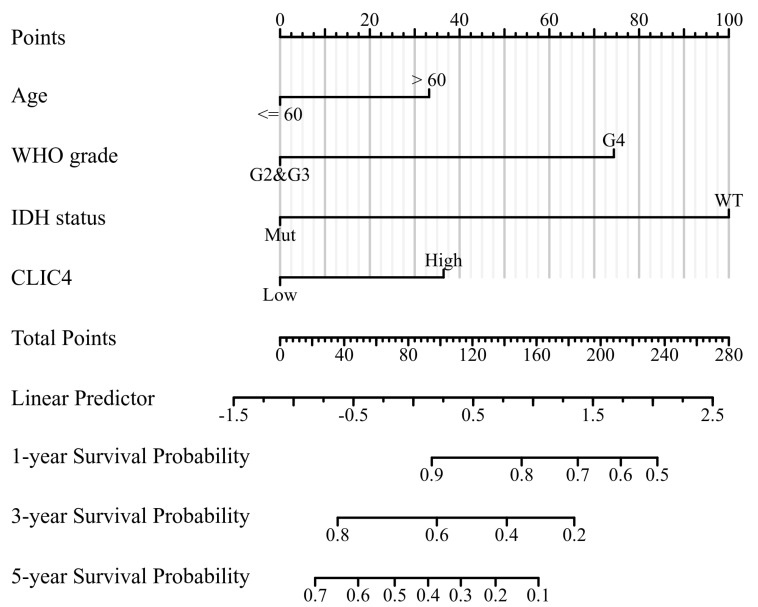
The nomogram model that includes clinicopathological factors (age, WHO grade, and IDH status) and *CLIC4* expression levels to predict the 1-, 3-, and 5-year survival rates of glioma patients.

**Figure 6 biomedicines-12-02579-f006:**
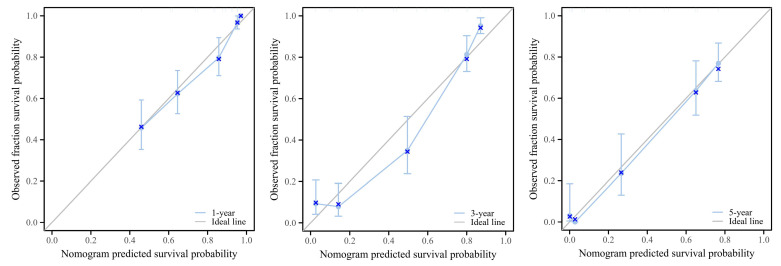
An analysis of the nomogram model’s calibration curve for predicting 1-, 3-, and 5-year survival rates.

**Figure 7 biomedicines-12-02579-f007:**
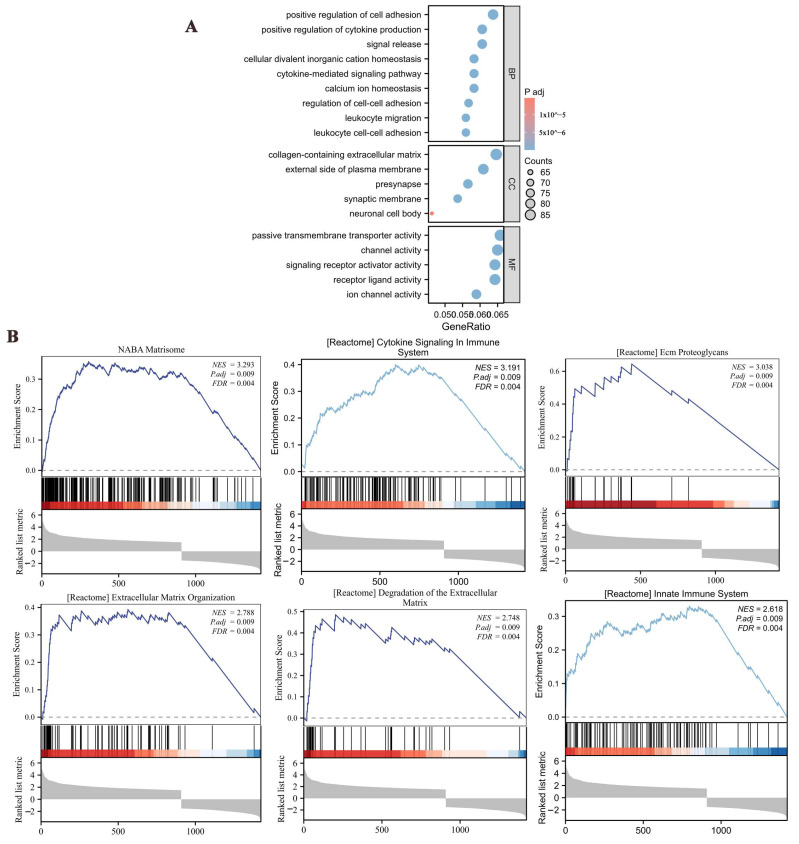
Based on the *CLIC4* expression levels in gliomas, functional enrichment analysis of differentially expressed genes (DEGs) was conducted. (**A**) GO enrichment analysis of the *CLIC4*-associated DEGs. (**B**) In glioma tissues, a Gene Set Enrichment Analysis was conducted using the *CLIC4*-associated DEGs between groups with high- and low-*CLIC4* expression.

**Figure 8 biomedicines-12-02579-f008:**
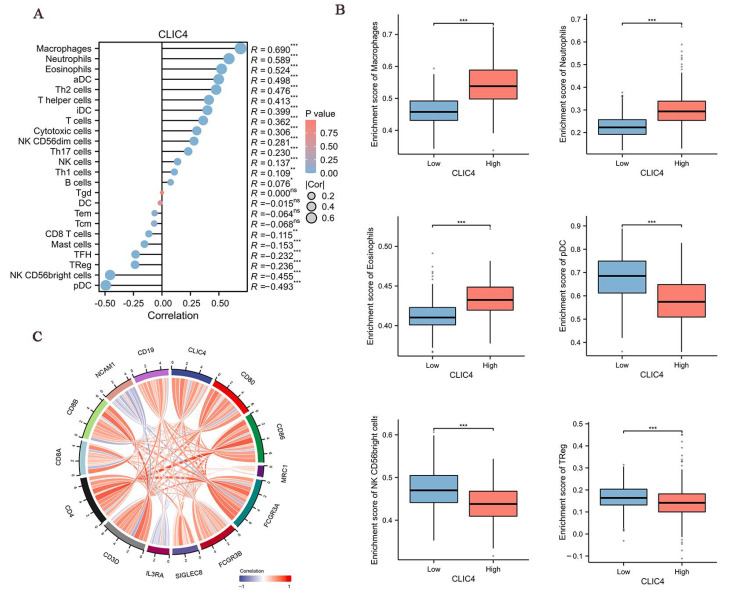
Inflammatory cells infiltrating into gliomas are correlated with *CLIC4* expression in cancer patients with *CLIC4* expression. (**A**) Spearman’s correlation analysis results in 24 immune cell types. (**B**) The infiltration levels of macrophage, neutrophils, eosinophils, pDCs, NK CD56bright cells, and treg cells. (**C**) The correlation chord diagram results between *CLIC4* expression and the immune cell markers. ns > 0.05 * *p* ≤ 0.05, ** *p* < 0.01, *** *p* < 0.001.

**Figure 9 biomedicines-12-02579-f009:**
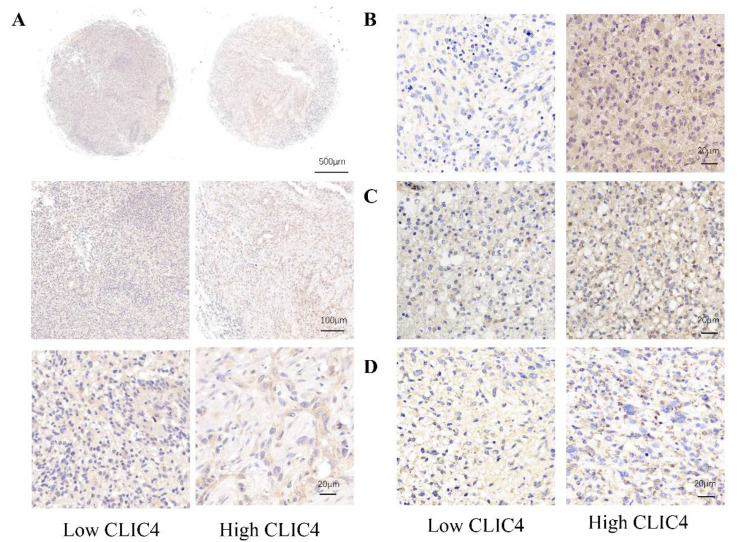
IHC staining of CD68 (**A**), VIM (**B**), SNAI1 (**C**), and MMP (**D**) in a low or high expression of CLIC4 glioma patients. Scale bars were 500 µm, 100 µm, and 20 µm.

**Figure 10 biomedicines-12-02579-f010:**
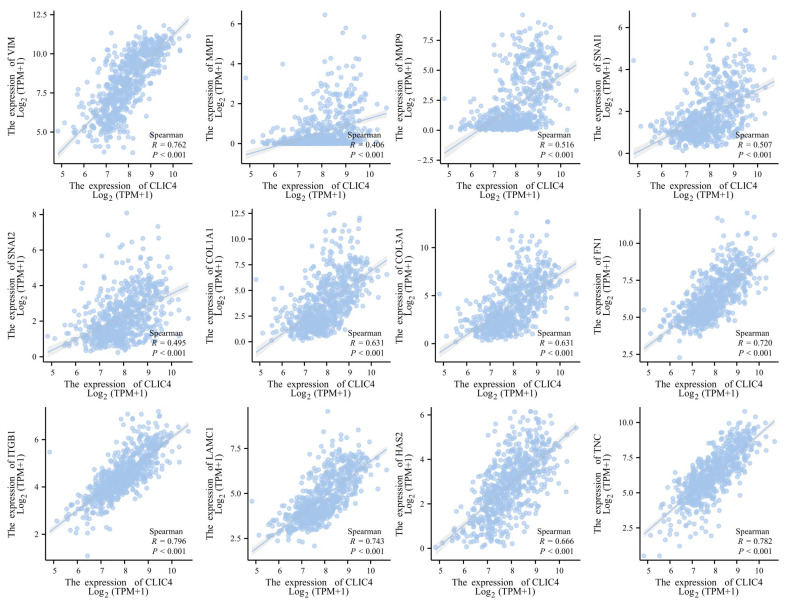
The correlation analysis between ECM-related genes and *CLIC4* expression in glioma patients.

**Table 1 biomedicines-12-02579-t001:** Clinicopathological characteristics, including *CLIC4* levels, were analyzed by Cox regression in glioma patients.

Characteristics	Total (N)	Univariate Analysis		Multivariate Analysis	
		Hazard Ratio (95% CI)	*p* Value	Hazard Ratio (95% CI)	*p* Value
Age	698				
≤60	555	Reference		Reference	
>60	143	4.696 (3.620–6.093)	<0.001	1.560 (1.154–2.108)	0.004
WHO grade	636				
G2&G3	468	Reference		Reference	
G4	168	9.538 (7.243–12.560)	<0.001	2.706 (1.893–3.867)	<0.001
IDH status	688				
WT	246	Reference		Reference	
Mut	442	0.116 (0.089–0.151)	<0.001	0.262 (0.182–0.379)	<0.001
CLIC4	698				
Low	348	Reference		Reference	
High	350	3.936 (2.997–5.168)	<0.001	1.628 (1.169–2.268)	0.004

## Data Availability

Data supporting this study’s findings are available from the corresponding author.

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
