# Peer review of "CLIC4 Is a New Biomarker for Glioma Prognosis"

_biomedicines, 2024, doi:10.3390/biomedicines12112579_

Round 1
Reviewer 1 Report
Comments and Suggestions for Authors
The manuscript by Zhichun Liu et al. entitled “CLIC4 is a new biomarker for glioma prognosis” is devoted to a systematic statistical analysis of the relationship between the expression of chloride intracellular channel 4 (CLIC4) and the malignancy and prognosis of gliomas, which is definitely substantiated and provides evidence that the use of CLIC4 expression level as an additional biomarker for the determination of gliomas can successfully complement existing diagnostic approaches. Although the applied analysis is not novel, it can be considered reliable, and the findings obtained are statistically significant.
The manuscript is recommended for publication in MDPI Biomedicines after consideration of the following comments:
Major:
1. Abstract is too general, some details on, e.g., high or low expression levels of CLIC4 are correlated with poor prognosis or, vice versa, indicated, low probability of glioma, should be added. Some more information could be added, e.g., highlighting the correlation between CLIC4 levels and GBM subtypes or specific genes. Brief conclusions in the Abstract should not state that the methods the authors used allowed them to reveal the correlation. The main value is the results themselves, what specifically was demonstrated, which relationship between features (and which features) is the most prominent.
2. Introduction: the authors mention that CLIC4 could be a target for cancer treatment, but the title states that CLIC4 is a biomarker for a prognosis, which is not discussed in the introduction. Therefore, the introduction, although being logical and clear, is inconsistent with the title. The authors need to add the corresponding emphases to exclude this inconsistency.
2. References to the public datasets need to be added: Line 54-56 “Cancer Genome Atlas-GBMLGG (699 samples), Chinese Glioma Genome Atlas-mRNAseq_693# (1,018 samples), Gill (92 samples), Gravendeel (284 samples), Murat (84 samples), Rembrandt (580 samples), LeeY (191 samples), GSE7696 (84 samples), and GSE50161 (130 samples).”
3. References to web-based tools such as GlioVis platform and Gene Expression Omnibus (GEO) need to be added.
4. Information on the origin of all the reagents needs to be added.
5. Please, add the reference to ImageJ. Although it is open-source software, it is still necessary to respect and evaluate the contribution of the creators.
6. Figure 1 should be reorganized (enlarged) to improve its perception.
7. Figures 2-6 need to be enlarged. The information is hardly distinguishable.
8. The subtitle “CLIC4 expression levels Correlate with the ECM in the glioma tissues” should be rephrased, since it is not clear what did the authors mean by saying that extracellular matrix correlate with CLIC4 expression levels. Probably, the ECM-related genes should be emphasized in it.
9. Lines 155-157 and 318-319 are almost identical, however, earlier mention lacks reference. Appropriate reference should be added to the lines 155-157 and one of the statements should be rephrased to avoid self-plagiarism.
10. The conclusions should be extended and reflect the main findings of the study. As already mentioned, the authors are invited to be precise in formulations, since „CLIC4 correlates with ECM“ is very unclear. The authors are suggested to make statements carefully. As it was already mentioned above, probably, the authors meant ECM-related genes and wanted to say that elevated expression of CLIC4 correlates with increased expression of ECM-related genes.
Minor:
1. Line 8: “MATHODS“ -> „METHODS“
2. Unnecessary highlighting in abstract
Author Response
- Abstract is too general, some details on, e.g., high or low expression levels of CLIC4 are correlated with poor prognosis or, vice versa, indicated, low probability of glioma, should be added. Some more information could be added, e.g., highlighting the correlation between CLIC4 levels and GBM subtypes or specific genes. Brief conclusions in the Abstract should not state that the methods the authors used allowed them to reveal the correlation. The main value is the results themselves, what specifically was demonstrated, which relationship between features (and which features) is the most prominent.
Response: We appreciate the reviewer’s insightful feedback on the abstract. To enhance clarity and focus, we have added specific details regarding the expression levels of CLIC4 and their correlation with glioma prognosis. Additionally, we have incorporated specific findings that highlight the DEGs (differentially expressed gene) of enrichment analysis. We have also revised the brief conclusion to emphasize the key findings rather than the methods used.
- Introduction: the authors mention that CLIC4 could be a target for cancer treatment, but the title states that CLIC4 is a biomarker for a prognosis, which is not discussed in the introduction. Therefore, the introduction, although being logical and clear, is inconsistent with the title. The authors need to add the corresponding emphases to exclude this inconsistency.
Response: We recognize that the emphasis on CLIC4 as a biomarker for prognosis was not sufficiently addressed, leading to an inconsistency with the title. To resolve this, we have revised the Introduction to discuss CLIC4's as a prognostic biomarker. This addition clarifies the research focus and aligns the Introduction with the title, providing a coherent foundation for the study's aims.
- References to the public datasets need to be added: Line 54-56 “Cancer Genome Atlas-GBMLGG (699 samples), Chinese Glioma Genome Atlas-mRNAseq_693# (1,018 samples), Gill (92 samples), Gravendeel (284 samples), Murat (84 samples), Rembrandt (580 samples), LeeY (191 samples), GSE7696 (84 samples), and GSE50161 (130 samples).”
Response: The references to the datasets have been added.
- References to web-based tools such as GlioVis platform and Gene Expression Omnibus (GEO) need to be added.
Response: The references to web-based tools such as GlioVis platform and GEO have been added.
- Information on the origin of all the reagents needs to be added.
Response: The information on the origin of all the reagents have been added
- Please, add the reference to ImageJ. Although it is open-source software, it is still necessary to respect and evaluate the contribution of the creators.
Response: The references to the imagej have been added.
- Figure 1 should be reorganized (enlarged) to improve its perception.
- Figures 2-6 need to be enlarged. The information is hardly distinguishable.
Response to 7 and 8: Figures 1-6 have been enlarged to make it distinguishable
- The subtitle “CLIC4 expression levels Correlate with the ECM in the glioma tissues” should be rephrased, since it is not clear what did the authors mean by saying that extracellular matrix correlate with CLIC4 expression levels. Probably, the ECM-related genes should be emphasized in it.
Response: To improve its accuracy, we have rephrased the subtitle to “CLIC4 expression levels Correlate with the ECM-related genes in the glioma tissues” This revised title emphasizes that the focus is on the relationship between CLIC4 expression levels and ECM-associated genes within glioma tissues, providing a clearer indication of the analysis conducted.
- Lines 155-157 and 318-319 are almost identical, however, earlier mention lacks reference. Appropriate reference should be added to the lines 155-157 and one of the statements should be rephrased to avoid self-plagiarism.
Response: We appreciate the reviewer’s attention to detail. We have now added the appropriate reference to the earliest mention. Additionally, to avoid redundancy and potential self-plagiarism, we have rephrased the latter statements to ensure that each mention of this information is distinct in its wording while preserving the original meaning.
- The conclusions should be extended and reflect the main findings of the study. As already mentioned, the authors are invited to be precise in formulations, since „CLIC4 correlates with ECM“ is very unclear. The authors are suggested to make statements carefully. As it was already mentioned above, probably, the authors meant ECM-related genes and wanted to say that elevated expression of CLIC4 correlates with increased expression of ECM-related genes
Response: Thank you for the constructive feedback on the Conclusions section. We have expanded the Conclusions to reflect the study's main findings more comprehensively. Specifically, we now precisely state that elevated CLIC4 expression correlates with increased expression of ECM-related genes and immune cell infiltration in glioma tissues. This adjustment clarifies the specific relationship observed, aligning the formulation with the study’s results and improving the accuracy of our statements.
Minor:1. Line 8: “MATHODS“ -> „METHODS“
Response: We apologize for this error, which has now been corrected in the manuscript.
- Unnecessary highlighting in abstract
Response: We apologize for the inconvenience. However, we did not encounter “the green highlighting” when opening the document using Microsoft Office LTSC Professional Plus 2021. Could you please confirm your Office version or check for other potential causes of this issue? We are available to assist in ensuring that the document displays correctly across different environments.
Reviewer 2 Report
Comments and Suggestions for Authors
This manuscript shows an interesting study and findings regarding that the CLIC4 expression potentially developed as a new biomarker for glioma prognosis. The manuscript is well written, and the methods and the results presentation were organized. Moreover, the conclusions are supported by the data presented. However, a couple of questions are required to be addressed before publication.
1. The number of references is relatively small, please add references. For the discussion of IDH, it is recommended to quote references: Int J Mol Sci. 2024 Aug 25;25(17):9226. doi: 10.3390/ijms25179226.
2. In this study, patients with IDH mutations and 1p/19q deletions exhibited lower CLIC4 expression. What is the expression of CLIC4 in other IDH mutant tumors (Cholangiocarcinoma)? This content should be discussed to increase the credibility of the results.
3. Paragraph composition in introduction section needs to be optimized. Please merge the short paragraphs.
Author Response
- The number of references is relatively small, please add references. For the discussion of IDH, it is recommended to quote references: Int J Mol Sci. 2024 Aug 25;25(17):9226. doi: 10.3390/ijms25179226.
Response: Thank you for your suggestion regarding the references. We have expanded the number of references in the manuscript to provide a more comprehensive context for our study. For the discussion on IDH, we have included the recommended reference from Int J Mol Sci (2024) to support our analysis. We appreciate this guidance and hope these additions enhance the depth and relevance of our discussion.
- In this study, patients with IDH mutations and 1p/19q deletions exhibited lower CLIC4 expression. What is the expression of CLIC4 in other IDH mutant tumors (Cholangiocarcinoma)? This content should be discussed to increase the credibility of the results.
Response:Thank you for your insightful suggestion. We have expanded the discussion to other IDH-mutant tumors, such as cholangiocarcinoma, to provide a broader context and enhance the credibility of our findings. Since this article focuses on CLIC4 and glioma, the relationship between CLIC4 and other IDH-mutant tumors will be explored in future studies.
- Paragraph composition in introduction section needs to be optimized. Please merge the short paragraphs.
Response: Thank you for your feedback regarding the paragraph composition in the introduction. We have revised the section by merging shorter paragraphs to improve flow and readability, optimizing the structure as suggested. We hope these changes enhance the clarity and coherence of the introduction.
Reviewer 3 Report
Comments and Suggestions for Authors In the current work, Liu and colleagues evaluated CLIC4, encoding chloride intracellular channel 4, as a novel potential biomarker for glioma aggressiveness. Using a variety of bioinformatic approaches and transcriptomic databases, the authors showed that glioma severity, as mediated by different glioma grades and key mutations (IDH, 1p19q), is significantly correlated with CLIC4 overexpression. Using correlation analysis, Liu et al. showed that high expression of CLIC4 is associated with glial-mesenchymal transition of glioma cells and poor prognosis in LGG and GBM patients, while CLIC4-related differentially expressed genes (DEGs) were enriched for functional terms related to regulation of extracellular matrix and immune response. Finally, the results obtained were verified in glioma patient samples and the correlation of high/low expression of CLIC4 with infiltration of tumour tissue with various immune cells was also found. In my opinion, the current manuscript is unfortunately not ready for publication in Biomedicines because of the superficial description of the data obtained and the lack of analysis of the correlation of CLIC4 overexpression with a number of other key glioma-associated regulators and processes. To make the manuscript more in-depth and understandable, the following major revisions are required: (1) Figure 3A. Problems with clinical samples. Please explain why the CLIC4 signal was significantly less pronounced in LGG samples than in NBT samples. (2) Fig. 3B. Why was such a small sample size chosen for NBT samples (only 4)? It is possible that biased results were obtained. Dear authors, please increase the number of samples in the NBT group. Also, please indicate in the figure whether there are significant differences between the NBT and LGG groups? (3) Figure 3C. Why is the CLIC4 signal in IDH WT LGG significantly higher than in the LGG samples shown in Fig. 3A?! (4) Lines 165-167 - Dear authors, please indicate the criterion you used to determine the strength of the correlation. In my opinion, ZEB1 expression does not correlate at all with CLIC4 expression (R = -0.09). Please check the data again and correct or comment. (5) Nomogram construction section. In my opinion, strong predictive accuracy is determined by WHO classification and IDH status, and CLIC4 expression is a less significant feature. Please construct a nomogram prediction model using other prognostic factors. (6) The article does not analyse in depth the conflicting results on immune infiltration. According to your data, low levels of CLIC4 are associated with a good prognosis in glioma patients (Fig. 4) and at the same time with high levels of pDCs and T-regs (Fig. 7A, B), which are known to be tumour-promoting immune cells. How can this be explained? (7) Fig. 8 - Unfortunately, the authors did not indicate why vimentin, snail and MMP1 were used for correlation analysis, while other very important participants of the ECM in glioma were ignored, such as fibronectin, collagens, slug, other matrix metalloproteinases. Please re-analyse and add these data to your article. (8) Discussion needs very serious revision. In its current state, the Discussion is just a listing of the data obtained without any in-depth analysis and comparison with published data. It remains completely unclear to what extent CLIC4's function as an ion channel determines its involvement in the processes identified by the authors. Please correct. (9) Lines 233-244. Dear authors, give more detailed information about the DEGs obtained (number, top 5 most up- and down-regulated). Minor comments: Line 15 - please decipher the abbreviation ECM Line 19 - please write Correlates in lower case. Line 21 - wrong sentence. How can gene expression be correlated with an object (ECM)? Correlation can only occur between dynamically changing processes. Line 57 and throughout the manuscript - all gene names, including CLIC4, should be in italics. Please correct throughout the manuscript. Line 74 - Authors have to specify which methods and software products were used to analyse the GEO datasets. Line 84 - Please provide details of all antibodies used. Line 102 - Snail1, Vimentin and Twist1 are protein names, not gene names. Please correct. Line 148 - Please delete |. Line 167 - Authors need to provide references as to why CHI3L1, RELB and TNFRSF1A were chosen as mesenchymal related genes. In my opinion, these are not classic genes of GMT. Line 176 - please delete the phrase 'patient progosis' as you have not yet done a survival analysis. Kaplan-Meier plots (Figure 4) are shown much lower. Line 186 and throughout the manuscript - please change um to µm. Line 188 - (a) please add space between WT and GBM; (b) please clarify where the 50 um and 500 um bars are. Tables 1,2 - Please replace <= with ≤. Figure 7 - in Circos plot, please keep only the curves starting from CLIC4. Correlation of immune cell markers between each other is unnecessary and redundant information. Line 273 - Please decode ns
Author Response
- Figure 3A. Problems with clinical samples. Please explain why the CLIC4 signal was significantly less pronounced in LGG samples than in NBT samples
Response: In fact, to correspond with Figures 1A and 1B, we conducted comparisons in Figure 3 between normal brain tissue and glioma, as well as between low-grade glioma (LGG) and glioblastoma (GBM), rather than directly comparing normal brain tissue with LGG. Additionally, based on the average optical density of IHC staining shown in Figure 3B, we can observe that some LGG samples have a relatively low average optical density (<0.35). However, due to the initial selection of unrepresentative IHC images for NBT and LGG, the result was not well supported. We have reselected IHC images to avoid this misunderstanding.
- 3B. Why was such a small sample size chosen for NBT samples (only 4)? It is possible that biased results were obtained. Dear authors, please increase the number of samples in the NBT group. Also, please indicate in the figure whether there are significant differences between the NBT and LGG groups?
Response: Our tissue microarray initially included only 4 normal tissue samples, so the previous manuscript analyzed only these four samples. To avoid potential bias, we have added 8 more normal tissue samples in this revision, which makes the result convincing. Additionally, we have included a comparison between NBT and LGG in figure 3B.
- Figure 3C. Why is the CLIC4 signal in IDH WT LGG significantly higher than in the LGG samples shown in Fig. 3A?!
Response: As mentioned above, the initial selection of unrepresentative IHC images for NBT and LGG led to this misunderstanding. We re-examined the immunohistochemistry results and are confident that the findings in our study are reliable.
- Lines 165-167 - Dear authors, please indicate the criterion you used to determine the strength of the correlation. In my opinion, ZEB1 expression does not correlate at all with CLIC4 expression (R = -0.09). Please check the data again and correct or comment.
Response: We apologize for our error. |r| > 0.90 indicates a Very high correlation; |r| ≥ 0.7 indicates a high correlation; 0.4 ≤ |r| < 0.7 indicates a moderate correlation; 0.2 ≤ |r| < 0.4 indicates a low correlation; |r| < 0.2 indicates a very low correlation. This should be a very low correlation, and we have revised the relevant sections in the text accordingly.
- Nomogram construction section. In my opinion, strong predictive accuracy is determined by WHO classification and IDH status, and CLIC4 expression is a less significant feature. Please construct a nomogram prediction model using other prognostic factors.
Response: Thank you for your insights regarding the nomogram construction. While we agree that WHO classification and IDH status are important prognostic factors for gliomas, to ensure the reliability and accuracy of the model predictions, these two factors are worth considering. Besides, our analysis suggests that CLIC4 expression also contributes valuable predictive information in this model, though it is not the most valuable.
- The article does not analyse in depth the conflicting results on immune infiltration. According to your data, low levels of CLIC4 are associated with a good prognosis in glioma patients (Fig. 4) and at the same time with high levels of pDCs and T-regs (Fig. 7A, B), which are known to be tumour-promoting immune cells. How can this be explained?
Response:Thank you for your valuable feedback. We have carefully considered your suggestion. We know that the results of bioinformatics analysis are not entirely reliable, and different datasets may yield varying outcomes. More accurate results await validation through cellular and animal experiments. In this study, we performed immunohistochemical staining for the macrophage marker (CD68) in glioma tissues with different CLIC4 expression levels, demonstrating a correlation between CLIC4 and CD68. Other contentious findings require validation in future studies.
- 8 - Unfortunately, the authors did not indicate why vimentin, snail and MMP1 were used for correlation analysis, while other very important participants of the ECM in glioma were ignored, such as fibronectin, collagens, slug, other matrix metalloproteinases. Please re-analyse and add these data to your article.
Response:We selected vimentin, Snail, and MMP1 as ECM-related genes due to their critical roles in promoting cell migration, invasion, and ECM remodeling in glioma. Vimentin and Snail are key markers of the epithelial-to-mesenchymal transition, while MMP1 is involved in ECM degradation, all of which are relevant to glioma progression and aggressiveness. However, we acknowledge the importance of including additional key ECM components in glioma. We re-analyse 9 more genes including MMP9,SNAIL2 , COL1A1, COL3A1 , FN1 , ITGB1, LAMC1,HAS2 and TNC.
- Discussion needs very serious revision. In its current state, the Discussion is just a listing of the data obtained without any in-depth analysis and comparison with published data. It remains completely unclear to what extent CLIC4's function as an ion channel determines its involvement in the processes identified by the authors. Please correct.
Response:Thank you for your constructive feedback on the Discussion section. We recognize the importance of providing an in-depth analysis and thorough comparison with existing literature. In our revisions, we integrate a more comprehensive discussion of our findings in the context of published research.
- Lines 233-244. Dear authors, give more detailed information about the DEGs obtained (number, top 5 most up- and down-regulated).
Response:Thank you for this helpful suggestion. We have revised the manuscript to include more detailed information about the DEGs, specifying the total number obtained as well as highlighting the top five most up-regulated and down-regulated genes. We hope this additional detail enhances the clarity and completeness of our findings.
Minor:
- Line 15 - please decipher the abbreviation ECM
Response:We have spelled out the abbreviation ECM for clarity.
- Line 19 - please write Correlates in lower case.
Response: “Correlates" has been corrected to lower case.
- Line 21 - wrong sentence. How can gene expression be correlated with an object (ECM)? Correlation can only occur between dynamically changing processes.
Response: We have rephrased this sentence to clarify the correlation between gene expression and ECM-related processes.
- Line 57 and throughout the manuscript - all gene names, including CLIC4, should be in italics. Please correct throughout the manuscript.
Response: All gene names are now in italics as required.
- Line 74 - Authors have to specify which methods and software products were used to analyse the GEO datasets.
Response: We have specified the methods and software products used to analyze the GEO datasets.
- Line 84 - Please provide details of all antibodies used.
Response: Details of all antibodies used have been added.
- Line 102 - Snail1, Vimentin and Twist1 are protein names, not gene names. Please correct.
Response: We have corrected Snail1, Vimentin, and Twist1 to gene names SNAI1, VIM, TWIST1
- Line 148 - Please delete |.
Response: The "|" has been deleted.
- Line 167 - Authors need to provide references as to why CHI3L1, RELB and TNFRSF1A were chosen as mesenchymal related genes. In my opinion, these are not classic genes of GMT.
Response: We have added references to support our choice of CHI3L1, RELB, and TNFRSF1A as mesenchymal-related genes i.e., (Liu J, Gao L, Ji B, et al. BCL7A as a novel prognostic biomarker for glioma patients. J Transl Med. Aug 6 2021;19(1):335. doi:10.1186/s12967-021-03003-0)
- Line 176 - please delete the phrase 'patient progosis' as you have not yet done a survival analysis. Kaplan-Meier plots (Figure 4) are shown much lower.
Response: The phrase "patient prognosis" has been removed.
- Line 186 and throughout the manuscript - please change um to µm.
Response: All instances of "um" have been changed to "µm."
- Line 188 - (a) please add space between WT and GBM; (b) please clarify where the 50 um and 500 um bars are.
Response: We have added space between "WT" and "GBM" and clarified the locations of the 50 µm and 500 µm scale bars.
- Tables 1,2 - Please replace <= with ≤.
Response: Tables 1 and 2: The symbols "<=" have been replaced with "≤."
- Figure 7 - in Circos plot, please keep only the curves starting from CLIC4. Correlation of immune cell markers between each other is unnecessary and redundant information.
- Thank you for your constructive feedback. Due to limitations in both time and capability, circos plot could not be revised at this time. We acknowledge the value of your suggestions and will consider these points in future studies to enhance the robustness of our findings. We appreciate your understanding and will continue to strive for rigor of our research.
- Line 273 - Please decode ns
15: "ns" has been added.
Reviewer 4 Report
Comments and Suggestions for Authors
In this study, the authors show that Chloride Intracellular Channel 4 (CLIC4) is linked to glioma malignancy and could thus constitute a potential prognostic biomarker of this disease. The results indicate that CLIC4 correlates with ECM and immune cell infiltration into glioma tissues.
The study is perfectly described in its methodology, a methodology which allows the authors to conclude their work with numerous and widely commented and discussed results. The quality of the article and its writing seem to me to allow non-specialists in the field a clear understanding and thus to reveal the interest of the study and especially the results.
It would be good if the authors developed the conclusion in order to further highlight the results obtained and to show the interest of their study.
Author Response
Response: Thank you very much for your positive and encouraging feedback. We appreciate your recognition of our methodology and the clarity of our results. In response to your suggestion, we have expanded the conclusion to further highlight our findings and emphasize the potential significance of CLIC4 as a prognostic biomarker in glioma. We hope this additional emphasis enhances the impact and accessibility of our study.
Round 2
Reviewer 3 Report
Comments and Suggestions for Authors
The authors have done a very good job with the manuscript and have corrected all of the problems I identified. I recommend this article for publication after correcting the following typo:
Line 289: please correct CYTKINE.
I thank the authors for their work and wish them success in their next research!
Author Response
Respond to Reviewer ’ comments in round 2
-
The authors have done a very good job with the manuscript and have corrected all of the problems I identified. I recommend this article for publication after correcting the following typo:
Line 289: please correct CYTKINE.
I thank the authors for their work and wish them success in their next research!
Response: Thank you for your positive feedback and for recommending our manuscript for publication. We are pleased to hear that our revisions have addressed your previous concerns. Besides, we apologize for the typo, which has now been corrected in the manuscript. The revised version is “KEGG_CYTOKINE _CYTOKINE_RECEPTOR_INTERACTION”.